# Calorimetry Technique for Observing the Evolution of Dispersed Droplets of Concentrated Water-in-Oil (W/O) Emulsion during Preparation, Storage and Destabilization

**Endarto Yudo Wardhono [1,\*], Mekro Permana Pinem [1,2] , Hadi Wahyudi [1] and Sri Agustina [1]**

[1] Chemical Engineering Department, University of Sultan Ageng Tirtayasa, Cilegon 42435, Banten, Indonesia; mekro-permana.pinem@utc.fr (M.P.P.); hadi.wahyudi@untirta.ac.id (H.W.); sriagustina@untirta.ac.id (S.A.)

[2] Integrated Transformations of Renewable Matter Laboratory (EA TIMR 4297 UTC-ESCOM), Sorbonne Universités, Université de Technologie de Compiègne, rue du Dr Schweitzer, 60200 Compiègne, France

\* Correspondence: endarto.wardhono@untirta.ac.id

**Abstract:** In this work, the evolution of dispersed droplets in a water-in-oil (W/O) emulsion during formation, storage, and destabilization was observed using a calorimetry technique. The emulsion was prepared by dispersing drop by drop an aqueous phase into an oil continuous phase at room temperature using a rotor-stator homogenizer. The evolution of droplets during (1) preparation; (2) storage; and (3) destabilization was observed using differential scanning calorimetry (DSC). The samples were gently cooled-down below its solid-liquid equilibrium temperature then heated back above the melting point to determine its freezing temperature. The energy released during the process was recorded in order to get information about the water droplet dispersion state. The mean droplet size distribution of the sample emulsion was correlated to its freezing temperature and the morphology was followed by optical microscopy. The results indicated that the calorimetry technique is so far a very good technique of characterization concentrated W/O emulsions.

**Keywords:** emulsion; droplets evolution; differential scanning calorimeter; freezing temperature; melting temperature.

---

## 1. Introduction

Emulsions are a disperse system, consisting of two immiscible liquids in which one of the liquids is dispersed in the other as droplets [1]. The liquid that makes up the droplets is referred to as the dispersed phase, and the surrounding is referred to as the continuous phase. The dispersed phase is generally the smaller fraction that presents in an emulsion system [2]. The type of emulsion depends on several parameters such as the nature of emulsifiers, the nature of the oil, and the effect of other materials [3]. In general, emulsions are classified according to the relative spatial distribution of the oil and aqueous phases [4]. Simple emulsions consist of a dispersed phase in a continuous phase. A system that consists of an oil phase dispersed in an aqueous phase is called an oil-in-water (O/W) emulsion, while a system that consists of a water phase dispersed in an oil phase is called a water-in-oil (W/O) emulsion. Mixed emulsions are obtained from two single emulsions of the same type that contain droplets of different compositions. These emulsions are oil-in-water ($O_1 + O_2$/W) emulsions or water-in-oil ($W_1 + W_2$/O) emulsions. Double emulsions are simple emulsions dispersed in a continuous phase. Double emulsions are complex systems where both W/O and O/W emulsions exist simultaneously. In the case of W/O/W emulsions, the oil droplets have smaller water droplets within them, and the oil droplets themselves are dispersed in a continuous water phase. The O/W/O

emulsions, however, consist of tiny oil droplets entrapped within larger water droplets, which in turn are dispersed in a continuous oil phase [4].

The process of converting oil and water phases to an emulsion system is called emulsification. The emulsion does not form spontaneously—energy input must be applied through shaking, stirring, or agitation [2]. The initial stage of emulsification is breakup and disrupt oil and water phases, so that one phase (of oil or water) becomes dispersed throughout the other. This stage is also called droplets disruption. The droplet disruption depends on the oil-water interfacial tension. The presence of surfactant decreases the interfacial tension and minimizes the effects of interfacial forces so that it facilitates the droplet disruption. Under mechanical agitation, the small droplets are constantly moving and the frequency of collision is very high. These lead to coalescence, then increasing their size, also called droplets coalescence. The presence of surfactant adsorbs to the surface of the droplets, forms a protective membrane to prevent the droplets from coming close together to coalesce. Rapid stabilization of the droplets against coalescence, once they are formed, depends on the time taken for the surfactant to be adsorbed to the surface of the droplets relative to the time between droplet–droplet collisions. This time depends on the flow profile that the droplets experience, as well as the nature of the surfactant used [5]. If the protective membrane between two droplets is broken, the droplets will merge, thereby reducing the total surface of the droplets. The emulsion tends to revert back to the initial state after it has been formed. The stability of the emulsion refers to the ability of an emulsion to keep its properties unchanged for a certain period—the more stable the emulsion, the more slowly its properties change [5]. To form a stable emulsion, it is necessary to ensure that the majority of the droplets fall within the same size ranges [6]. The size of the droplets produced during preparation is important because it determines the stability and rheological characteristics of the emulsions. An emulsion may become unstable due to a number of different types of physical and chemical processes. Physical instability results in an alteration in the spatial distribution, whereas chemical instability results in an alteration in the kind of molecules present. Creaming, sedimentation, flocculation, coalescence, phase inversion, and Ostwald ripening are examples of mechanisms that initiate physical instability. Oxidation and hydrolysis are common examples of creating chemical instability [2].

A concentrated emulsion is a class of emulsions, which is characterized by a minimum internal phase volume fraction of 0.74 [7,8]. It is generally applied for food, agrochemical, and pharmaceutical products. In this work, the concentrated water-in-oil (W/O) emulsion is developed for crop protection purposes. The emulsion that was formulated contains a certain amount of polysaccharides in the internal aqueous phase which was dispersed into the vegetable oil continuous phase. In its application, polysaccharides serve a thickening adjuvant of the water-based spray solution which reduces the drift of the spray [9]. For commercial reasons, the emulsion designed has to present long stability and destabilize immediately into the water dilution system during spraying application. In our previous works [10,11], we have shown that it was possible to get such W/O emulsion containing the high internal aqueous phase (75% v/v). The combination of polyglycerol polyricinoleate (PGPR) as surfactant and glycerol co-surfactant in the vegetable oil continuous phase was beneficial to retain the emulsion without any phase inversion. During the water dilution process, the destabilization was intentionally created to release the entrapped polysaccharides from the primary W/O emulsion by introducing a certain amount of chemical agents [12].

In understanding the basics of the emulsion developed from the droplets formation point of view, the objective of this study is to track the evolution of the droplets during preparation (formation), storage (stability), and destabilization (deformation) over time. The way the droplets evolve with time was observed by differential scanning calorimetry (DSC) along with laser diffraction granulometry and optical microscopy as the control of the observations. Many techniques have been used to characterize the emulsion which are generally based on the analysis of the droplets size distribution. However, the majority of these measurements require a dilution of the sample. It may create perturbation on the surface properties of the droplets and on interactions between the droplets, inducing flocculation or

even coalescence phenomena [13]. For the emulsion system, DSC detects the freezing and melting phenomena of the dispersed phase and the continuous phase during a heating or cooling program. It monitors the energies involved including the heat exchanges between the sample and a reference either versus time at a constant temperature or versus temperature. These give information on the droplets dispersed state without any dilution of the system [14]. Hence, this work shows that calorimetry provides a simple, fast and precise technique to characterize the droplet size distribution of a concentrated (W/O) emulsion.

## 2. Materials and Methods

### 2.1. Materials

The aqueous phase (w/w): 3.5% of polysaccharides Carboxy Methyl Cellulose (CMC) from Roeper GmbH, Hamburg, Germany. 10% of glycerol from Thermo Fischer Scientific, Illkirch, France. Demineralized water (conductivity of 0.06 mScm$^{-1}$) produced by a purification chain, Veolia, France, was used for all experiments.

Oil phase (w/w): 10% of surfactant polyglycerol polyricinoleate (PGPR) with Hydrophilic-Lipophilic Balance (HLB) number of 4 and vegetable oil, rapeseed oil was supplied by Mosselman, Ghlin, Belgium.

Chemical agent: Cynthiorex PMH 1125, destabilizer (ethoxylated fatty alcohol) with HLB number of 6 from Mosselman, Ghlin, Belgium.

### 2.2. Methods

#### 2.2.1. Sample Preparations

W/O emulsion preparation was carried based on the work of Wardhono et al., 2014 [11]. Samples were prepared at room temperature (25 °C) using a rotor-stator (POLYTRON PT-3100D-Kinematica, Kinematica AG, Luzern, Swiss). 25% v/v of oil phase were poured previously into a vial and then the rotor speed was set up at 3000 rpm. 75% v/v of the aqueous phase were then introduced dropwise into the oil phase in 5 min. The shearing rate was then increased gradually (1000 rpm per 30 s) up to 14,000 rpm and kept at this speed for 15 min.

Storage, accelerated aging tests were carried out to predict the shelf life of the emulsions. The tests were conducted in accordance with the methods recommended by the Collaborative International Pesticides Analytical Council (CIPAC) [15] for liquid formulations devoted to phytosanitary use. The test involves extrapolations from higher to lower temperatures and from shorter to longer storage periods as shown in Table 1.

**Table 1.** Storage temperature and duration of stability test, under copyright permission from [15].

| Temperature | Duration of Stability Trial |
|---|---|
| 54 °C | 14 days |
| 45 °C | 6 weeks |
| 35 °C | 84 days |

The fresh emulsion was filled into a graduated tube and stored at a controlled temperature oven at 54 °C for 14 days. The samples which stay stable during the test are estimated equivalent to the store for 2 years at room temperature.

Destabilization was performed according to the work of Wardhono et al., 2016 [12]. An amount of 1 mL of chemical agent was dropped into a beaker glass containing 10 mL of primary emulsion then mixed for 2 min using a magnetic stirrer at a gentle speed (200 rpm, at room temperature). The addition of chemical agents into emulsion was intended to speed up the droplet's deformation of emulsion.

### 2.2.2. Characterization

DSC analysis was performed using DSC131-Evo Setaram (Caluire, France). An amount of 30 mg of the sample was dropped into a sample cell then inserted into a pan holder. Temperature program for the measurement was adjusted as follows: (1) started with equilibrated isotherm temperature at 30 °C for 600 s; (2) cooled down the temperature from 30 to −60 °C with cooling rate 3 °C/min; (3) set the second isotherm temperature at −60 °C for 300 s; and (4) heated back to the initial temperature at 30 °C with heating rate 3 °C/min. The temperature programs were supervised by software Calisto.

Laser Diffraction Granulometry, the droplet size was measured by laser particle size Mastersizer X (Malvern Worcestershire, England). One drop of the emulsion was diluted into a 100 mL of oil solution with a flow of 1 L/min at room temperature. The results are presented in terms of the distribution size in the volume base because for calorimetric measurements it is the energy involved during freezing/melting depending on the mass, which is recorded. The mean droplet size is characterized by

$$D_{43} = \frac{\sum N_i D_i^4}{\sum N_i D_i^3} \tag{1}$$

where $N_i$ is the number of droplets in size range i and $D_i$ is the middle diameter of size range i.

Optical Microscopy, the morphology of the sample was observed using optical microscope NIKON Labophot-2. The sample object was observed under high power of 100× and the microscope image captured was copied by software IrfanView 64 bit version (Nikon, Tokyo, Japan).

## 3. Results and Discussions

### 3.1. DSC Thermogram Interpretation

DSC is a technique to measure the thermal properties of materials to establish a connection between temperature absorbed or emitted during controlled heating, and the specific physical properties of the substance. It is the only method for direct determination of enthalpy associated with the process of interest [16,17]. The principle of the DSC is based on the measurement of heat flux required for the temperature of the sample increases or decreases. This heat flux is directly proportional to the heat capacity of the material for a given temperature. The amount of heat involved depends on whether the process is exothermic or endothermic. By observing the difference in energy flow between the sample and reference, the thermal properties of materials can be measured [18]. DSC is widely used for the decomposition behavior determination of the polymer [19–22]. In the emulsion system, DSC monitors the solidification process inside the droplets and calculates the temperature dependence of the nucleation rate [23]. The sample is regularly cooled below and heated above to its solid-liquid equilibrium temperature to determine the freezing, $T_f$ and melting temperature, $T_m$. The energies involved are recorded as peaks presented in exothermic and endothermic flows versus temperature. The droplet sizes and the way they evolved with time might be determined from the peaks obtained [14].

A DSC thermogram of a freezing water droplet is presented in Figure 1. The straight line in the figure indicated there is no water droplet that has been frozen before $T_1$. As the temperature is lower than 0 °C, the ice formation starts to form inside the droplet even in a very small number because the nucleation rate is nearly zero [23]. The nucleation rate is drastically increasing between temperature $T_1$ and $T_2$. All the droplets have full frozen as $T_2$ is attained, but it depends on the sensibility of the calorimeter used. The exothermic peak (Figure 1, ① is the energy released and the area below the baseline (Figure 1, ②) is the energy to heat of the sample [24]. The $T^*$ refers to the mean freezing temperature because the maximum number of droplets are frozen in this temperature [25]. According to Dalmazzone, 2009 [13] the $T^*$ may be correlated to the mean diameter of dispersed droplets because the maximum number of droplets are expected to freeze around this temperature.

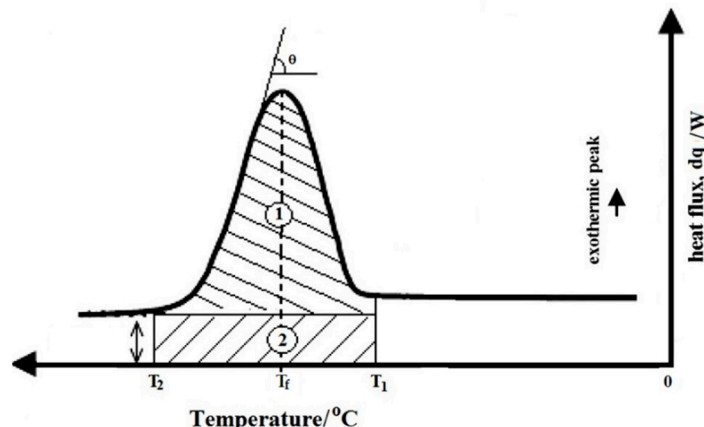

**Figure 1.** Differential scanning calorimetry (DSC) thermogram of energy released inside of water droplet.

Figure 2 shows the thermogram of a bulk aqueous phase (in dotted line) and a sample W/O emulsion (in solid line). During the cooling process from 30 to −60 °C, water freezes at different temperatures depending on their size. For 30 mg of the sample aqueous solution with diameter, $\emptyset$ = 1 mm, a mean freezing temperature, T* is found at −14 °C (Figure 2, *i*). For 30 mg of the sample W/O emulsion with a mean droplet size distribution, $D_{43}$ = 4 μm, T* is found −39 °C (Figure 2, *ii*), while the oil continuous phase freezes at T* = −50 °C (Figure 2, *iii*). During the heating process, the temperature returns to the initial one, all the water melts at the same temperature, namely $T_m$ = 0 °C, as seen in Figure 2, *v* for the water dispersed droplets of sample emulsion and Figure 1B, *vi* for the bulk aqueous phase. And the melting temperature of the oil continuous phase is found at $T_m$ = −33 °C (Figure 2, *iv*). According to Clausse et al., 2014 [23], freezing is the result of nucleation phenomena, temperature variations can induce a change in morphology, consequently, all the water droplets do not freeze at the equilibrium point, so their freezing temperatures are scattered.

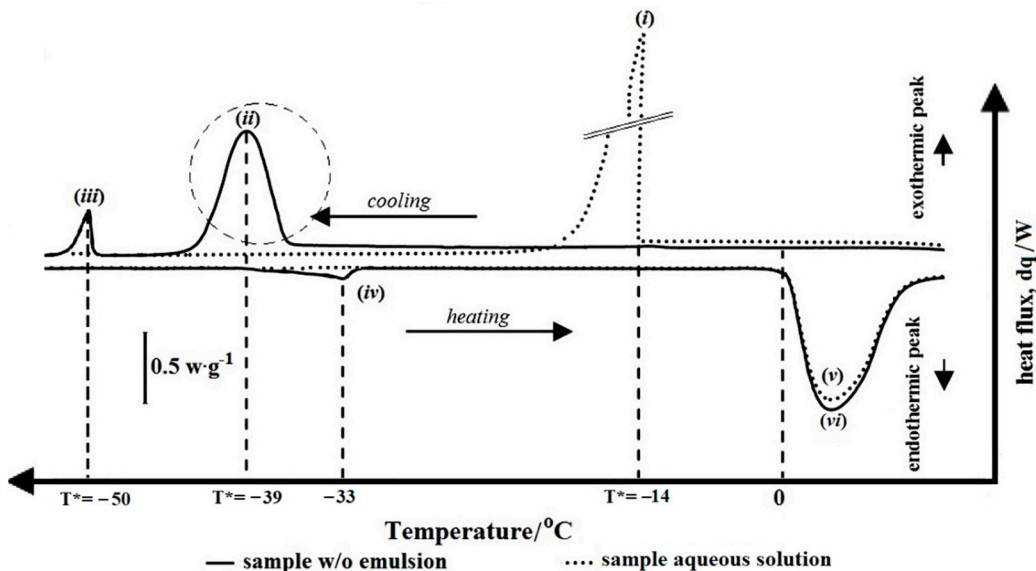

**Figure 2.** DSC thermogram of exothermic and endothermic peak of sample w/o emulsion and sample aqueous solution.

The freezing and melting of the sample references in accordance with their masses are summarized in Table 2.

The shape of the exothermic peak can be used to characterize of emulsion to be analysed. An O/W emulsion typically has the same peak in Figure 2, *i* due to the presence of water in the continuous

phase. A broken one is analogous to the one in Figure 2, *i* too. Otherwise, a W/O emulsion has similar to Figure 2, *ii*, the Gaussian-shaped indicates that a homogenous dispersed droplet population was formed. The energy released is scattering around to the individual mean freezing point. While for a heterogeneous one, the exothermic peak displays more than one peak with an asymmetrical shape [25].

**Table 2.** Freezing and melting temperature of the samples.

| Sample | Size | | T* (°C) | T$_m$ (°C) |
| --- | --- | --- | --- | --- |
| | **Mass** | **Diameter, Ø** | | |
| Sample aqueous solution | 30 mg | 1 mm | −14 | 0 |
| Sample w/o emulsion | 30 mg | | | |
| - dispersed droplet | | 4 µm | −39 | 0 |
| - oil phase | | - | −50 | −33 |

*3.2. Droplets Size Evolution*

3.2.1. Droplets Formation during Preparation

Droplets Disruption

To create a stable emulsion, the droplets have to form within the small range sizes, which are achieved by mechanical agitation. Intensity and duration of agitation depend on the time required to homogenize uniformly ingredients. The droplets formation are then produced into two steps: (1) droplet disruption; and (2) droplet coalescence [2]. In this work, droplets disruption was carried out by mixing two phases at which the aqueous phase was introduced drop by drop into the oil phase at the shearing rate of 3000 rpm in 5 min. The rate was then increased step by step up to 14,000 rpm in 5 min. The results of droplets formed during the disruption process are shown in Figure 3.

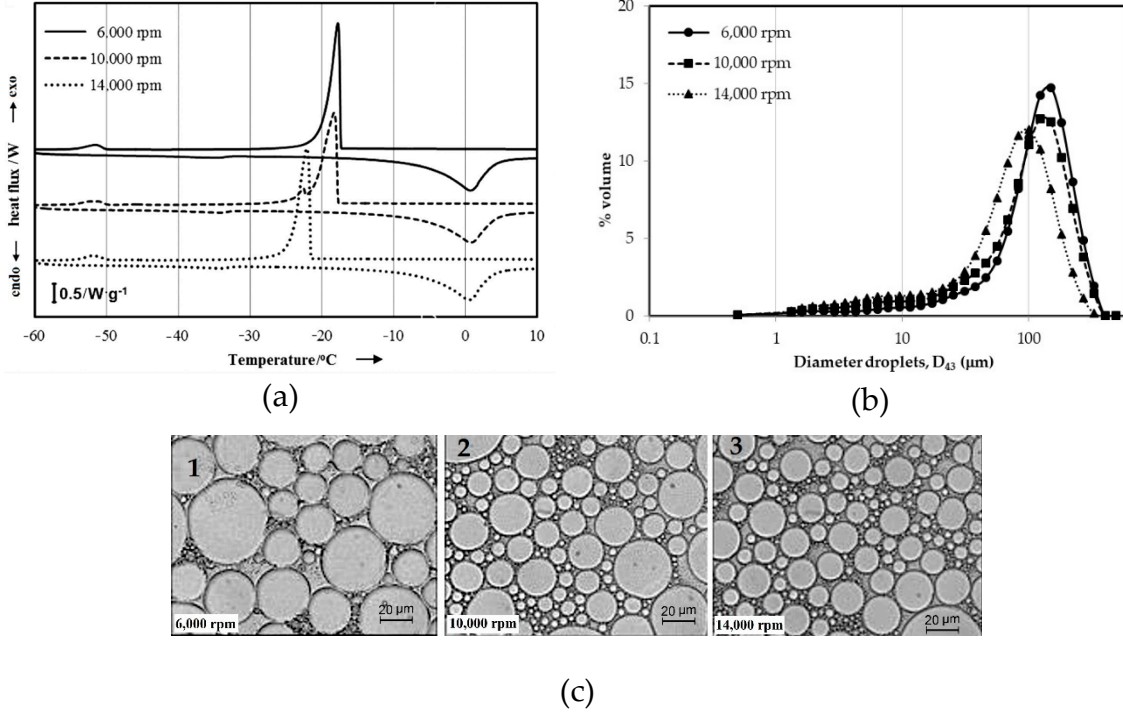

**Figure 3.** Observation droplets formation during droplets disruption by (**a**) DSC test; (**b**) laser diffraction granulometry; and (**c**) optical microscopy.

Summary of the droplets formation during the disruption process is presented in Table 3.

**Table 3.** Formation of the droplet sizes during droplets disruption and droplets coalescence.

| | Droplets Disruption (rpm) | | | | Droplets Coalescence (min) | | | |
|---|---|---|---|---|---|---|---|---|
| | 3000 | 6000 | 10,000 | 14,000 | 0 | 5 | 10 | 15 |
| $T^*$ (°C) | - | −15.4 | −15.8 | −21.3 | −21.3 | −23.4 | −33.3 | −35.6 |
| $D_{43}$ (µm) | - | 488 | 451 | 153 | 153 | 102 | 15 | 10 |

At the lower speed, 3000 rpm, the water and oil phases are disrupted and intermingled. By increasing the shearing speed to 6000 rpm, the droplets started to form. The freezing temperature was found at $T^* = -15.4$ °C (Figure 3a, solid line), however the exothermic peak is still located around the bulk aqueous phase. The shape of the peak is sharp which implies that a large amount of water has frozen at the same time, so the energy is released in a very short time. In line with the increasing of the shearing rate to 10,000 rpm, the $T^*$ moves to the lower temperature −15.8 °C (Figure 3a, dashed line) with the tail shifted along with the lower temperature. It reflects the droplets not being completely emulsified yet. As the rate attains to 14,000 rpm (Figure 3a, dotted line), the disruption leads the larger droplets to the smaller ones. Two peaks are found in the thermogram, namely $T^* = -21.3$ °C and $T^* = -27$ °C, which indicate that the polydispersed population of the droplets with the presence of large droplets dominated the emulsion system. The peak at −51 °C is the solidification of the oil phase. The volume mean droplet size distribution, $D_{43}$ decreased step by step from $D_{43} = 488$ µm at 6000 rpm to $D_{43} = 451$ µm at 10,000 rpm then to $D_{43} = 153$ µm at 14,000 rpm. The microscopic observation shows that the distribution of the droplets is in a heterogeneous population, but a small population of large droplets begin to present as the speed increased.

Droplets Coalescence

The droplets coalescence was conducted by maintaining the shearing rate at 14,000 rpm for 15 min. The samples were observed in every 5 min. The results are reported in Figure 4.

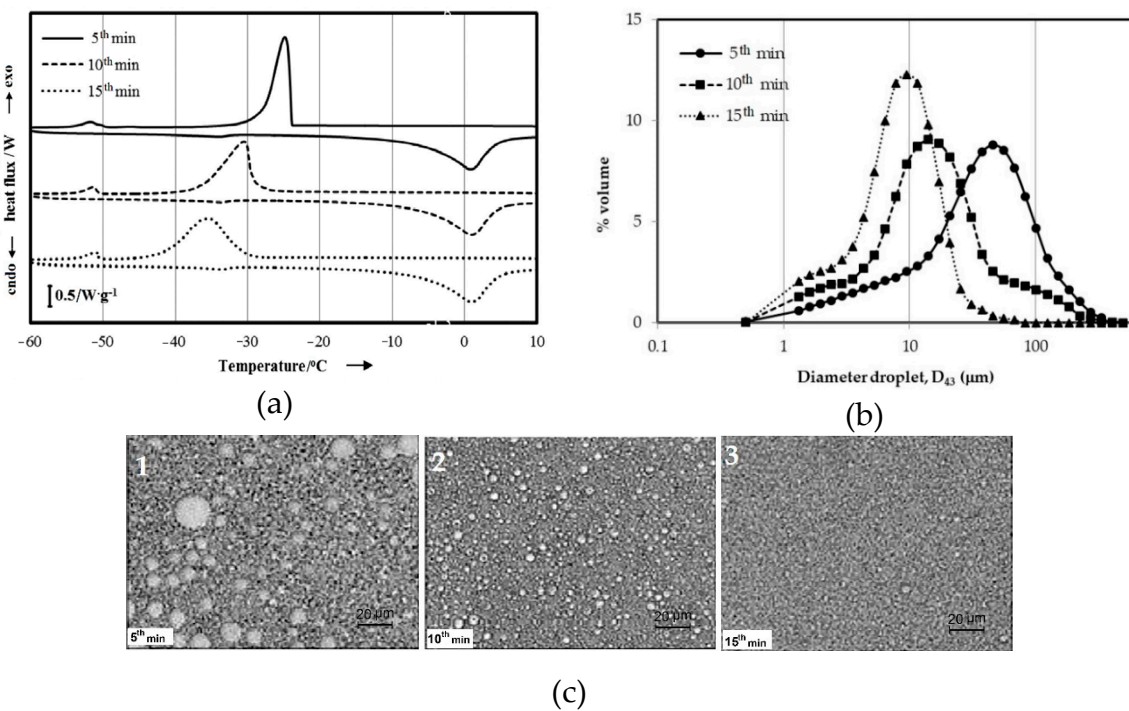

**Figure 4.** Observation droplets formation during droplets coalescence by (**a**) DSC test; (**b**) laser diffraction granulometry; and (**c**) optical microscopy.

The thermograms reveal that in the first 5 min of stirring, T* = −23.4 °C (Figure 4a, solid line). An asymmetrical signal correlates to the polydispersed population of the droplets. This result corresponds to the microscopic observation at which big droplets displayed in some parts of the image (Figure 4c, 1). The size of the mean droplet is $D_{43}$ = 102 μm. For the next 5 min, T* shifts to the lower temperature of −33.3 °C (Figure 4a, dashed line), the peak is slightly similar to the one in Figure 2, *ii*. The image shows that the big droplets disappeared then spotted in a smaller one to fluctuate randomly in size (Figure 4c, 2). The mean size of these droplets is 15 μm. After 15 min, T* is found −35.6 °C (Figure 4a, dotted line), the peak is a bell shape that implies emulsion with the monodisperse distribution. This is in favor of the optimum time for the coalescence of the droplets. This result is in agreement with optical microscopy. The image shows that the droplets are in the homogeneous population (Figure 4c, 3) with the minimum $D_{43}$ attained being 10 μm.

### 3.2.2. Droplets Shelf-Life during Storage

To determine the droplets shelf-life during the storage, an accelerated aging test was carried at elevated temperatures. The test is intended to increase the rate of chemical degradation or physical change of the products in a relatively short time. According to Dobrat, 1995 [15], the resistance emulsion-based products against phase separation before and after the test are equivalent to the stability for 2 years under normal conditions. The shelf-life of the droplets during the storage is presented in Figure 5.

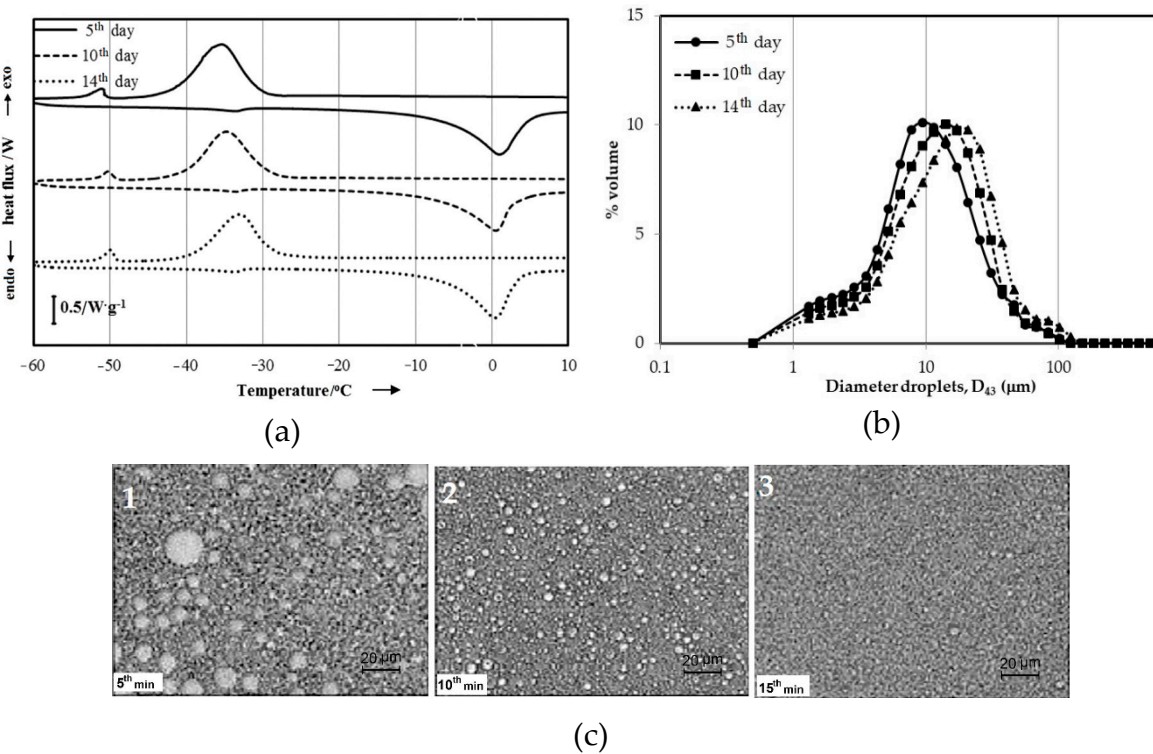

**Figure 5.** Observation droplet shelf-life during storage by (**a**) DSC test; (**b**) laser diffraction granulometry; and (**c**) optical microscopy.

After 5 days of storage (Figure 5a, solid line), the exothermic peak still has the same shape as the fresh one before storage (Figure 5a, dotted line), where the mean freezing temperature is still around T* = −35.5 °C. T* moves to the higher temperature −34.8 °C (Figure 5a, dashed line) after 10 days then shifts to −33.8 °C (Figure 5a, dotted line) after 14 days. The droplets beginning to coalesce during the test is indicated by increasing T* to the higher temperature. Even though the exothermic signal is the same as the fresh one, the droplets merge continuously but they did not break at all.

These results are in accordance with the droplet size calculation that is summarized in Table 4. The emulsions resist destabilization during storage. A similar volume means droplet size distribution is obtained for the emulsions in 0 days and 5 days, $D_{43} = 10$ µm. The droplets size, $D_{43}$ at the 10 days is 11 µm and then it increases to 14 µm at the end of the test. Figure 5c shows the population is the mono-disperse distribution for all samples of the test. It means that the emulsion produces long term stability. In general, temperature affects emulsion stability significantly. Temperature changes the properties of interfacial films and surfactant solubility in oil and water phases. Temperature increases the frequency of drop collisions. It also reduces the interfacial viscosity, which results in faster film drainage rate and faster droplets coalescence [26].

**Table 4.** Droplet shelf-life during storage.

| | Storage, 54 °C | | | |
|---|---|---|---|---|
| | **0 Day** | **5th Day** | **10th Day** | **14th Day** |
| T* (°C) | −35.5 | −35.5 | −34.8 | −33.8 |
| $D_{43}$ (µm) | 10 | 10 | 11 | 14 |

### 3.2.3. Droplets Deformation during Destabilization

Basically, destabilization of emulsion involves at least three steps process, namely flocculation, coalescence, and finally separation of the oil and water phase. In this work droplets deformation was carried out intentionally by addition chemical agent, to speed up destabilization. The observation of droplets deformation as a function of chemical agent concentration which was introduced into the emulsion system is presented in Figure 6 and is summarized in Table 5.

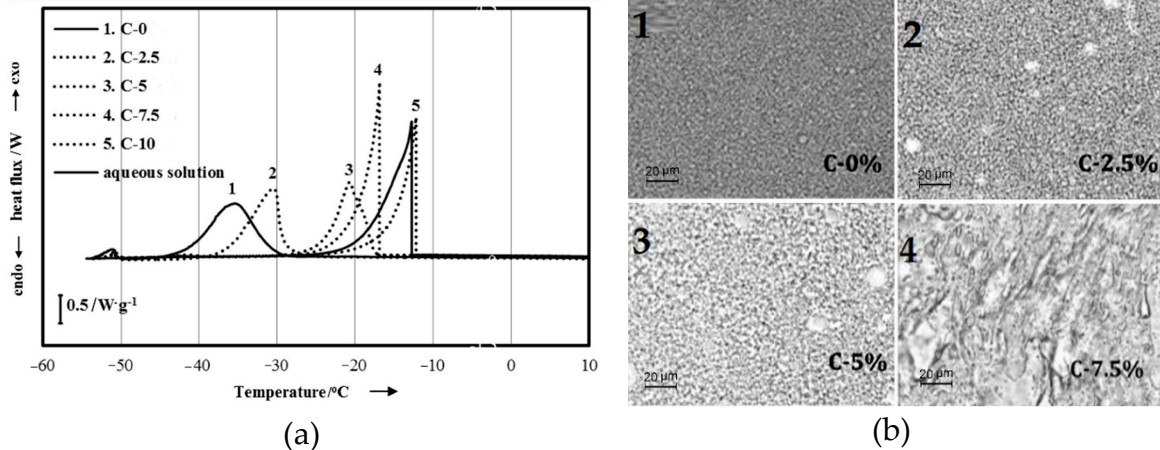

(a)  (b)

**Figure 6.** Observation droplet deformation during destabilization by; (**a**) DSC test; (**b**) microscopy.

**Table 5.** Droplets deformation during destabilization.

| | Chemical Agent (% w/w) | | | | |
|---|---|---|---|---|---|
| | **0** | **2.5** | **5** | **7.5** | **10** |
| T* (°C) | −35.5 | −30 | −21 | −17 | −13 |
| $D_{43}$ (µm) | 10 | 28 | 163 | - | - |

The DSC test shows the displacement T* to the higher temperatures from −35.6 to −30 °C for the sample C-2.5 and from −35.5 to −21 °C for the sample C-5 indicates that the population of the singles droplets has coalesced to form a large size due to the presence of the chemical agent. These results are in line with the microscopic observation where a bigger population of large droplets present in some part of the sample containing ≤ 5% w/w of chemical agent compare to the one in 2.5% w/w. The droplet

size measurement is found to be about 28 μm and 163 μm for the samples, respectively C-2.5 and C-5. For the samples containing 7.5% and 10% w/w of chemical agent, the shape of the signals observed is totally changed (Figure 6a, C7.5 and C10) in a form like that of the aqueous solution. It implies that the emulsion has totally broken. These results were then confirmed using optical microscopy observation (Figure 6b, 4) were presents that dispersed droplets population in the emulsion system has disappeared.

## 4. Conclusions

DSC is a calorimetry technique that can be applied to follow the formation, stability, and deformation of the dispersed droplet of complex emulsion. It is a simple, fast and precise technique and only requires small amounts of samples. The test is easy to set up and consists of submitting a sample into a regular cooling and heating program without any dilution system. The DSC thermogram gives reliable information on the droplets dispersed state, which helps the user to interpret the characteristics of different types of emulsion systems. DSC is also suitable to get information about the composition of dispersed and continuous phases as well as the polydispersity of the emulsion. However, it is always recommended to use at least two different techniques in order to comprehend the whole process that occurred in the emulsion system.

**Author Contributions:** E.Y.W. took care about Sections 1–4, proposed the subject of the review to all the other authors and took care about the general planning of the work and supervised all the work; M.P.P. organized Section 2; S.A prepared the figures and tables and organized references; H.W. wrote the abstract and the conclusions and Finally, all the authors contributed equally to the general organization of the manuscript and its revision, with helpful suggestions about the content and the style of the text.

**Funding:** This work was founded by Ministry of Research, Technology, and Higher Education of the Republic of Indonesia (Kemenristek DIKTI) and Universitas Sultan Agung Tirtayasa (UNTIRTA) under the scheme *Hibah Penelitian Dasar* with the contract number: 7/E/KPT/2019.

**Conflicts of Interest:** The authors declare no conflict of interest.

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
