# Peer review of "Calorimetry Technique for Observing the Evolution of Dispersed Droplets of Concentrated Water-in-Oil (W/O) Emulsion during Preparation, Storage and Destabilization"

_applsci, doi:10.3390/app9245271_

Round 1

Reviewer 1 Report

This study is very interesting. However, this manuscript should be revised before publication.

This manuscript type is “Article”, not “Communications”. The authors should enrich the Introduction. For instance, physical and/or chemical studies of W/O emulsion should be shown. In addition, because this journal is “Applied Sciences”, application of W/O emulsion should be shown. In Materials subsection, the authors described the “PGPR” (line 55, page 2). Is this “Polyglycerol polyricinoleate”? The full names of chemical compounds should be described. In lines 109 and 110, the authors described “According to Dalmazzone, 2009 [4] the T* may be correlated to the mean diameter of dispersed droplets.” This point is very important for this manuscript. However, there are a little explanation. The authors should show the information in detail. In conclusion section, the authors should show what they clarified. They did not describe about physical and chemical properties of W/O emulsion shown in this study.

Author Response

The responses are attached

Reviewer 2 Report

The authors have written a manuscript entitled “Calorimetry technique for observing the evolution of dispersed droplets of a water-in-oil, W/O emulsion during formulation, storage and destabilization”.

The manuscript needs extensive English editing which makes it quite difficult to fully understand and review the content of this manuscript. There are often sentences that are not comprehensible in their present form. 

Additionally, there are already several studies that use DSC to study the morphological characteristics of emulsions, and thus the authors need to clearly state what has already been done and what is the novelty of this research.

Also, Figures and Tables' captions are not always complete and the references in the captions are not always uniform with the way they are mentioned in the manuscript. This should also be addressed.

Some microscope images do not have enough quality and the images should contain a scale bar.

There are other issues that need major revision and the attached manuscript contains some suggestions, indications and doubts addressed to the authors.

In the current form this manuscript should not be published.

Author Response

The response is attached

Reviewer 3 Report

The comments for authors  are presented in the attached file

Author Response

The response is attached

Reviewer 4 Report

P1 L29-30; this is not correct. You can have W/O emulsion with very high water content. We prepared W/O emulsions with as low as 10% lipid (oil phase) and 90% water. Please read our paper on this: Kulkarni, C.V., R. Mezzenga, and O. Glatter* (2010) Soft Matter, 6: 5615-5624. We also prepared O/W emulsions where oil phase was 5% and aqueous phase was 95%. See this paper: Kulkarni, C. V.*, V. Vishwapathi, A. Quarshie, Z. Moinuddin, J. Page, P. Kendrekar and S. Mashele. (2017) Langmuir, 33 (38), pp 9907–9915. P1 L33-34: Static light scattering (SLS) technique with special conditions can be used to determine the droplet size without actually diluting the sample. See this paper: Kulkarni, C.V., R. Mezzenga, and O. Glatter* (2010) Soft Matter, 6: 5615-5624. Be consistent when using abbreviations and their long names. Some places you use brackets “()” while at some places you use comma “,” to separate these. What is PGPR? Write long names at their first occurrence. P2 L63: use the Standard English number format: 14,000 or 14.000? Also check other numbers in the manuscript. What was the temperature for formulation? Maintained constant or prepared at Room Temperature? P2 L69: name the chemical agent and detailed conditions (e.g. conc. solvent name, proportion) to be able to reproduce this work P2 L71: “deformation”: do you mean destabilization or breaking of droplets? P3 L108: put “]” for the reference 17. P4 L128: what is the theoretical basis for this statement? How can Gaussian shape relate to the homogeneous dispersion of droplets? Figure 2: if the new droplets are forming then what is getting disrupted? Do you mean: size reduction of emulsion droplets by mechanical agitation? Novelty, conclusions and prospects (or application to the relevant scientific community) are not explicit.

Author Response

The response is attached

Round 2

Reviewer 2 Report

The authors made a clear effort to improve the quality of the paper, particularly in terms of the introduction and also in terms of the English editing.

Nonetheless, there are still some errors in the manuscript that need correction, such as the sentence that was added to show the novelty of this article, at the end of the introduction section.

Some other suggestions are in the manuscript attached.

The conclusion could be clearer.

Also, there are several tables in the manuscript that are called Table 1. This needs to be corrected. Please verify all the tables and figures numbers, as well as the way they are referenced along the manuscript.

Author Response

Thank you very much

Reviewer 3 Report

The authors completed the requested corrections

Author Response

Thank you very much

Reviewer 4 Report

Most of my comments have been answered but not all.

Author Response

Thank you very much